# Treatment of Metastatic or High-Risk Solid Cancer Patients by Targeting the Immune System and/or Tumor Burden: Six Cases Reports

**DOI:** 10.3390/ijms20235986

**Published:** 2019-11-28

**Authors:** Andrea Nicolini, Paola Ferrari, Riccardo Morganti, Angelo Carpi

**Affiliations:** 1Department of Oncology, Transplantation and New technologies in Medicine, University of Pisa, 56100 Pisa, Italy; paolaferrari2266@libero.it; 2Section of Statistics, University Hospital of Pisa, 56100 Pisa, Italy; r.morganti@ao-pisa.toscana.it; 3Department of Clinical and Experimental Medicine, University of Pisa, 56100 Pisa, Italy; angelo.carpi@med.unipi.it

**Keywords:** advanced solid cancer, tumor burden, minimal residual disease, immune evasion, immunotherapy

## Abstract

This article summarizes the histories of six patients with different solid tumors treated with a new strategy based on tumor burden reduction and immune evasion as potential targets. All six patients were at a high risk of relapse and were likely to have a minimal residual disease following conventional therapy: biochemical recurrence (BCR) following radical prostatectomy (RP) (two prostate cancers patients), removal of distant metastases (one colorectal and one breast cancer), and complete response (CR) of distant metastases to conventional therapy (one breast cancer and one esophageal–gastric junction cancer). Four of the patients, two after RP and BCR, one after removal of a single pulmonary metastasis from breast cancer, and one after CR to chemotherapy of peritoneal metastases and ascites from an esophageal–gastric junction primary cancer, regularly received cycles of a new drug schedule with the aim of inhibiting immune suppression (IT). In these four patients, preliminary laboratory tests of peripheral blood suggested an interleukin (IL)-2/IL-12 mediated stimulation of cellular immune response with a concomitant decrease in vascular endothelial growth factor (VEGF) immune suppression. The fifth case was a breast cancer patient with distant metastases in CR, while receiving beta-interferon and interleukin-2 in addition to conventional hormone therapy. To date, all five patients are alive and doing well and they have been unexpectedly disease-free for 201 and 78 months following BCR, 28 months following the removal of a single pulmonary metastases, 32 months following CR to chemotherapy of peritoneal metastases and ascites, and 140 months following diagnosis of multiple bone metastases, respectively. The sixth patient, who had colorectal cancer and multiple synchronous liver metastases and underwent nine surgical interventions for metastatic disease, although not disease-free, is doing well 98 months after primary surgery. Our six cases reports can be interpreted with the hypothesis that immune manipulation and/or a concomitant low tumor burden favored their clinical outcome.

## 1. Introduction

Recently, we proposed a refined version of the patho-biological model of advanced solid cancers reported by Hanahan and Weisberg [1], and we hypothesized that a close relationship exists between tumor growth and immune evasion [2]. Following experimental and clinical observations [2], we proposed that an instrumentally undetectable tumor or the prolonged quiescent/non-growing state of residual cancer cells is associated with a better immune response and more efficacious active immune manipulation compared with a detectable and/or growing tumor. The aim of this article is to summarize further advances that support these concepts and to discuss several clinical cases treated accordingly.

## 2. Six Cases Reports and Methods

### 2.1. Cases

All six cases were at a high risk of a detectable relapse owing to biochemical recurrence (BCR) after radical prostatectomy or radical prostatectomy in addition to successful radiotherapy of local recurrence (*n* = 2, prostate cancer), an undetectable residual disease following surgical removal of distant metastases (*n* = 2, one with endocrine-resistant breast cancer and the other with colorectal cancer), or complete response (CR) of metastatic disease to systemic therapy (*n* = 2, one with breast cancer and the other with gastric-esophageal junction cancer). Four of these patients (two with prostate cancers and BCR, one with endocrine-resistant breast cancer, and one with gastric–esophageal junction cancer) were given the below-described new schedule of immunotherapy (IT), which is based on the repurposing of non-anticancer agents. These were the first four consecutive cancer patients at a high risk of relapse who came to our observation after we had set up the IT schedule. Consistent with our refined model [2] and our previous report [3], the patient with endocrine-resistant breast cancer and the patient with metastatic gastric–esophageal junction cancer also had to regularly receive additional chemotherapy (ChT) cycles. However, the latter did not receive it because of a lack of pathological confirmation of peritoneal metastases. Another patient with metastatic breast cancer took part in a pilot study that we have reported on several times [4,5,6,7,8]. According to the protocol, she cyclically received a different immunotherapy schedule with beta-interferon and interleukin-2 (IL-2) in addition to continuous conventional hormone salvage therapy. Before they were treated with IT, these five patients signed a witnessed written informed consent form. The sixth patient, who had colorectal cancer and synchronous liver metastases, is reported here for his prolonged survival following an unusually high number (nine) of surgical removals of metastatic disease. All six patients were submitted to an intense complete clinical-radiological assessment (every 2–4 months) and serum tumor markers’ monitoring (every 1–3 months) consistent with the involved pathology: prostate specific antigen (PSA) for prostate cancer (two patients), carcinoembryonic antigen (CEA)-tissue polypeptide antigen (TPA)-carbohydrate antigen 15.3 (CA15.3) for breast cancer (two patients), CEA- carbohydrate antigen 19.9 (CA19.9) for colon-rectal (one patient) and gastric-esophageal junction cancer (one patient). The last observation took place on 30 October 2019.

### 2.2. The Rationale for a New Immunotherapy (IT) Strategy

#### 2.2.1. Low Dose Cyclophosphamide (CY)

Experimental studies carried out in mice showed that the administration of 100 mg/kg CY by i.v. as a single dose combined with immune cells allowed for the complete and permanent regression of methylcolantrene-induced fibrosarcoma by eliminating tumor-induced suppressor T cells [9]. Additionally, 30 mg/kg CY administered by as a single i.p. injection depleted CD4+CD25+ T cells in tumor-bearing animals induced by tolerogenic cell clones isolated from a rat colon carcinoma and injected into syngeneic hosts. CY delayed tumor growth and cured rats bearing established tolerogenic tumors when followed by an immunotherapy, which had no curative effect when given alone [10]. In other clinical studies, most of which were conducted in patients with metastatic melanoma, CD8+T suppressor activity was impaired after 300 mg/m^2^ CY was given alone by rapid i.v. infusion as a single bolus [11], or cyclically administered every four weeks followed by an autologous melanoma cell vaccine three days later [12], or followed by an intra-dermal injection of autologous irradiated melanoma cells mixed with Bacillus Calmette–Guerin [13]. The highest decrease in T suppressor cells occurred 19 days after the single CY bolus and 49–105 days after its cyclical administration [12]. In a more recent clinical study [14], CD4+ FOXP3+ regulatory T cell depletion by low-dose CY prevented recurrence in patients with large condyloma acuminate after laser therapy. Patients received 50 mg of CY orally once a day for one week. These patients did not show any signs of recurrence in the first six weeks, although nine recurrences were observed thereafter. Interestingly, following another week of low-dose CY treatment alone, seven out of nine patients resolved completely without second recurrence.

#### 2.2.2. Low Dose Dexamethasone (DX)

Glucocorticoids, at pharmacological doses, are the most widely used immunosuppressive agents in clinical medicine. In one experimental study [15], tumor-bearing mice that were treated with a single low-dose i.p. injection of DX (0.75 mg/kg) showed a significantly decrease in all systemic inflammatory parameters, and the state of immunological unresponsiveness to “established” immunogenic tumors (known as an “immunological eclipse” in the presence of tumor antigens) was reversed. The reversal was not sufficient to inhibit the primary growing tumors, however, the injection of dendritic cells (DC) loaded with tumor antigens after inoculation with DX was associated with significant inhibition of the growth of both “established tumors” and remnant tumor cells after tumor excision. Low-dose DX likely inhibited the accumulation of immature myeloid precursors and polymorphonuclear (PMN) Gr1+Mac1+cells in the blood, the spleen, and the tumor site in which tumor growth is favored as they differentiate into highly immune-suppressive tumor-associated macrophages (TAMs).

#### 2.2.3. Celecoxib (Cyclooxygenase 2 (COX2)-Inhibitor)

In an experimental study [16], positive feedback between prostaglandin E (2) (PGE (2)) and COX2 redirected the differentiation of human dendritic cells toward stable myeloid-derived suppressor cells (MDSC). Conversely, the disruption of COX2 PGE (2) feedback by COX2 inhibitors suppressed the production of MDSC-associated suppressive factors and the cytotoxic T lymphocytes (CTL)-inhibitory function of fully developed MDSCs from cancer patients. In another study [17], N-[2-(Cyclohexyloxy)-4-nitrophenyl]methanesulfonamide (NS-398), a selective COX2 inhibitor, suppressed the invasiveness of oral squamous cell carcinoma cell lines by down-regulating matrix metalloproteinase-2 production and activation. It was also reported [18] that adaptive T reg cells expressed COX2 and suppressed T cells in a PGE (2)-cyclic adenosine monophosphate (cAMP)-dependent manner, which could be reversed by COX2 inhibitors. In the first [19] of two further studies, the TAM phenotype was changed from M2 (pro-tumor) to M1 (anti-tumor) by celecoxib, and it was concomitant with the up-regulation of the M1-related interferon–gamma cytokine. In the second study [20], celecoxib inhibited systemic PGE (2) production and MDSC development and accumulation in the tumor microenvironment (TME). In the first of two clinical trials in which advanced colorectal cancer patients [21] received 200 mg of celecoxib capsules orally (given twice daily for no less than eight weeks), a significantly improved three-year survival rate was observed. The second study was a randomized, double-blind, placebo-controlled trial [22], in which 400 mg of celecoxib was orally taken twice daily for six months. This lung cancer chemoprevention study recruited former smokers who had not had cancer or who had stage 1 non-small-cell lung cancer (NSCLC) treated with curative resection. Baseline and follow-up bronchoscopies (the first after six months and the second at one year) were carried out at predetermined sites. A significant decrease in the Ki-67 labeling index, which was the primary endpoint, was observed after six months of treatment with celecoxib.

#### 2.2.4. Retinyl Palmitate Plus DL-Alpha Tocopheryl Acetate (Vitamins A Plus E)

After intestinal absorption of vitamin A, an oxygenation process at the cytoplasmatic level, generated all-trans-retinoic-acid (ATRA) and 13-cis retinoic acid, which are the main active metabolites of vitamin A. Inhibiting carcinogenesis by controlling both cellular differentiation and proliferation [23,24,25] is a well-known property of retinoids; for this reason, they have often been proposed for cancer chemoprevention [26]. Retinoids have also been reported to be powerful inhibitors of neo-angiogenesis and to synergize with IL-2 to decrease serum vascular endothelial growth factor (VEGF) [27]. In addition, immune-modulatory properties have been attributed to retinoids. In particular, in experimental studies, ATRA improved the differentiation of myeloid cells and the immune response in cancer patients [28], and 13-cis-retinoic acid and beta-carotene increased both the number of IL-2 receptors and the percentage of peripheral blood lymphoid cells that express surface markers for T-helper cells [29]. Other experimental studies showed that ATRA eliminated immature myeloid cells in tumor-bearing mice [30] and that physiological concentrations of retinoic acid favored the development of myeloid dendritic cells over granulocytes in cultures of bone marrow cells from mice [31]. Immature myeloid suppressor cells (GR1+CD115+) mediate the development of tumor- induced T regulatory cells and T-cell anergy in tumor-bearing hosts [32]. In some pilot clinical trials, in which most patients with different solid cancers were in CR or PR after conventional therapy, 13-cis retinoic acid at 0.5 mg/kg body weight/day was orally administered in addition to IL-2 as maintenance therapy. In these patients, overall (OS) and/or progression-free survival (PFS) significantly improved [27,33]. As beta-carotene and as a retinoid, Dl alpha tocopheryl (Vit E) acts as an anti-oxidant [34] and exerts anti-tumoral action, thus contrasting tumor growth [35,36,37]. Moreover, the anti-tumoral action of Dl alpha tocopheryl is synergistic with that of some common antiblastics [38,39,40]. The below reported IT schedule is consistent with these experimental and clinical findings. It was developed to inhibit immune-suppression induced by MDSCs (DX, celecoxib) and Treg cells (CY), in addition to carcinogenesis and tumor growth (vitamins A plus E). In accordance with our refined model, this IT schedule was given in the context of an undetectable minimal residual disease or low metastatic tumor burden.

#### 2.2.5. A New IT Schedule

Four patients (cases 1–4) received the following new IT schedule. On the first day of the first week (day 1, week 1), the patients were given 1.5 mg of dexamethasone in solution twice a day by mouth, and one CY pill at 50 mg/day on days 1–7 (days 1–7, week 1). In the following week (days 8–14; week 2), they received four (two in the morning and two in the evening) 200 mg celecoxib pills. In weeks 3–4, the patients received 30,000 IU of retinyl palmitate (vitamin A) plus DL-alpha tocopherol acetate (vitamin E) at 70 mg per pill, three pills a day (in the morning, at midday, and in the evening). A cycle of IT lasted for four weeks (weeks 1–4) and was followed by two weeks (weeks 5–6) of rest. In the patients receiving only the new IT cycles (three patients) or the previously reported IT cycles combined with hormone-therapy (one patient), treatment had to be continuously administered for at least five years or until clinical-radiological progression. In the patient who received the new IT and ChT cycles, each new IT cycle was followed by the ChT cycle, and the cycles were continuously given until clinical-radiological progression or for five years, as follows: four consecutive cycles of IT (4 cycles, 24 weeks each followed by 1 month of rest) in seven months were repeated six times (42 months), followed by four cycles of ChT in three months repeated six times (18 months) (Table 1). 

### 2.3. Principal Immunological Parameters, Cytokines, and Growth Factors in the Four Patients Receiving Immune-Suppression-Inhibiting Therapy

In each of the four patients who received the same immune-suppression-inhibiting therapy, the following were measured in peripheral blood a few days before immunotherapy (basal values) and compared with the values determined during the last two weeks of immunotherapy: the total number of lymphocytes and CD3+, CD4+, CD8+, CD4+25+ (T reg), CD8+25+, and CD4+R0 (memory cells) T subpopulations; CD16+56+ (NK) cells and CD19+ B lymphocytes; interleukin (IL)-1beta, IL-1ra (receptor), sIL-2, sIL-2R, IL-6, IL-8, and IL-10; monocyte chemo-attractant protein-1 (MCP-1) and tumor necrosis factor (TNF) alpha cytokines, as well as tumor growth factor (TGF)-beta1, fibroblast growth factor (FGF), epidermal growth factor (EGF), insulin-like growth factor-1(IGF-1), platelet-derived growth factor (PDGF), and VEGF-A. All cytokines and growth factors were assayed by a quantitative immunosorbent assay. In the serum of healthy donors, the cut-off values were as follows: <8.7 pg/mL, <18.3 pg/mL, <8.5 pg/mL, <8.9 pg/mL, <590 pg/mL, <14 pg/mL, and <44.3 ng/mL for IL-1beta, IL-1ra, IL-6, IL-10, IL-12, TNF alpha, and TGF-beta1, respectively; serum sIL-2 was not detectable; for sIL-2R, the limit of detection was 0.21 ng/mL; serum IL-8 ranged from not detectable to 666 pg/mL; for MCP-1, the minimal detectable dose (MDD) ranged from 0.57 to 10 pg/mL and 1.7 pg/mL was the mean of MDD; for FGF, the MDD ranged from 0.01 to 0.07 pg/mL and the mean of MDD was 0.03 pg/mL; for EGF, the MDD was <0.7 pg/mL; for IGF-1, the serum values ranged from 46 to 363 ng/mL and 172 ng/mL was the mean; for PDGF, the MDD ranged from 0.9 to 5.9 pg/mL and the mean of MDD was 1.7 pg/mL; and finally, for VEGF-A, the limit of detection was 7.9 pg/mL and the mean of the detectable samples ranged from 45.7 to 144.3 pg/mL. Within and between assay coefficient of variation was 7.6%, 5%, 5.8%, 7.7%, 4.2%, 8.1%, 4.8%, 5.5%, 6.5%, 6%, 6%, 5%, 3%, 7%, 3%, and 6.2%, respectively, and 10.3%, 9%, 7.4%, 10%, 8.2%, 10.8%, 10.6%, 11.5%, 12.6%, 5%, 5.7%, 7%, 5%, 13%, 7%, and 4.3%, respectively. The following commercial kits were used: eBioscience GmbH (Wien, Austria) for IL-1beta, sIL-2, sIL-2R, IL-6, IL-8, IL-10, and VEGF-A; R and D systems (Abington, UK) for IL-1ra, IL-12, TNF alpha, MCP-1, FGF, and EGF; and DRG Instruments, GmbH (Marburg, Germany) for TGF-beta1, IGF-1, and PDGF. Continuous data are described by the mean (sd) and median (range). Because of the few available determinations (the evaluation is ongoing), statistical analysis of the difference in the serum values of all measured parameters was carried out using both the unpaired Student’s *t*-test (two-tailed) and the Mann–Whitney test (two tailed). Differences with *p* < 0.05 were considered significant.

### 2.4. Patient 1

#### 2.4.1. History

On 9 October 2002, a 57-year-old male underwent nerve-sparing radical prostatectomy (RP), orchidectomy, vesiculectomy, and bilateral lymphadenectomy for prostate adenocarcinoma. The pre-operative serum PSA was >30 ng/mL. pT3a No Mo was the post-operative TNM classification, G2, 4-5 Gleason, stage III. On 27 January 2003, biochemical recurrence (BCR) (two consecutive values higher than 0.2 ng/mL) was detected [41,42] with a serum PSA level of 0.60 ng/mL. On 1 April 2003, a PSA level of 1.17 ng/mL was found. From 19 April 2003 to 3 June 2012, intermittent androgen deprivation therapy (IAD) was administered with 250 mg flutamide pills, three pills/day, and an luteinizing hormone-releasing hormone (LHRH) agonist was given subcutaneously once every four weeks. From 3 June 2012 to 12 October 2013, because of increasing serum PSA levels (on 29 May 2012, a PSA of 2.24 ng/mL was found), flutamide was replaced with 50 mg bicalutamide, one pill/day, and an LHRH agonist was given subcutaneously once every four weeks. Because PSA levels continued to increase (on 7 October 2013, a PSA value of 9.85 ng/mL was found), from 12 October 2013 to 15 April 2014, bicalutamide was replaced by 300 mg of cyproterone acetate, which was administered i.m. once a week, and an LHRH agonist was given subcutaneously once every four weeks. A trans-rectal ultrasound on 4 November 2013 and whole body (WB) computed tomography (CT) on 30 November 2013 detected a 27 × 26 mm nodule adherent to the bladder wall, and it was suspected to be local recurrence; this suspicion was confirmed by biopsy. On 17 December 2013, the serum PSA level was 18.28 ng/mL. From 18 April 2014 to 18 June 2014, he received external beam radiotherapy (EBRT; 7000 cGray in total). Following the fractioned EBRT, the serum PSA levels continuously decreased and, on 5 May 2016, the PSA value was 0.24 ng/mL. On 1 June 2016, the patient started cyclic IT, which was consistently administered according to the above-reported schedule. The serum PSA levels further and progressively decreased. On 27 August 2018, 50 months after EBRT, serum PSA was undetectable, and then it fluctuated between 0 and 0.01 ng/mL. The last PSA observation on 5 August 2019 was 0.01 ng/mL.

#### 2.4.2. Comment

A study [43] recruited 35 prostate cancer patients submitted to RP. Ten of these patients were pT2a-b N0M0 and 24 pT3a-b N0M0. All 35 patients with a macroscopic histologically-proven local recurrence received 30 Gy (21 patients) or 40 Gy (14 patients) of intensity-modulated brachytherapy (IMBT), combined with 30 Gy of EBRT. In the subgroup of 21 patients, after a mean follow-up of 29 months, PSA progression-free survival (PFS) was 24%, and the mean time interval to a PSA increase was 16 months. In the other subgroup of 14 patients, the mean follow-up was 26 months, the PSA progression-free survival (PFS) was 43%, and the mean time interval to a PSA increase was 10 months. Rising PSA values of <2 to 4 ng/mL have been reported to be predictive factors for successful radiotherapy of local recurrences after RP [44,45]. In the above-mentioned study, the mean rising PSA prior to salvage radiotherapy was 5.02 ng/mL, while it was 18.28 ng/mL in our reported patient. Moreover, the serum PSA nadir has been reported in the literature to occur 12–24 months after radiotherapy (RT) [46]; in our patient, it was reached after 50 months and maintained at 62 months.

### 2.5. Patient 2

#### 2.5.1. History

On 22 April 2010, a 65-year-old male underwent RP and vesiculectomy for prostate adeno-carcinoma. The pre-operative serum PSA was 13.66 ng/mL. pT3b No Mo was the post-operative TNM classification, G3, 4 + 4 Gleason, stage III. On 7 July 2010, the serum PSA was 0.01 ng/mL. On 10 April 2013, three months after RP, BCR was found to have occurred with two consecutive serum PSA values higher than 0.2 ng/mL (PSA level of 0.23 ng/mL). From 22 April 2013 to 10 May 2015, he received IAD therapy with 150 mg bicalutamide pills (once a day) and, on 9 February 2015, the serum PSA level was 0.02 ng/mL. From 11 May 2015 to 1 November 2015, no adjuvant therapy was given and, on 6 October 2015, the serum PSA level was 0.27 ng/mL. From 2 November 2015 to 29 October 2017, cyclic IT was administered. On 5 October 2017, the serum PSA level was 0.65 ng/mL. From 30 October 2017 to 13 May 2018, he received only 150 mg bicalutamide pills (once a day) and, on 9 May 18, the serum PSA level was 0.24 ng/mL. After that, as a result of a misunderstanding, the patient delayed the beginning of cyclic IT from 14 May 2018 to 11 June 2018. Moreover, during the first cycle of this new round of cyclic IT (from 11 June 2018 to 7 July 2018), he took one instead of four celecoxib pills a day. On 11 July 2018, the serum PSA level was 0.58 ng/mL. From 22 July 2018, he correctly underwent IT cycles and, on 11 May 19, the serum PSA was 0.99 ng/mL. He started on 50 mg bicalutamide pills (once a day), and an LHRH agonist was given subcutaneously once every three months. On 10 August 19, PSA was undetectable; therefore, he again started the IT schedule. On 10 October 19, the last PSA determination was 0.04 ng/mL.

#### 2.5.2. Comment

In a study [47], researchers examined 148 patients who prospectively enrolled in sequential Institutional Review Board approved vaccine protocols. The T stage was T1–T2 in 54% and T3–T4 in the remaining 46%; Gleason was 6–7 in 71% and 8–9 in the remaining 29%. All subjects had recurrent disease and no detectable radiological metastases. Recurrent disease manifested as BCR after surgery or as clinically localized disease at the sites for which they had undergone radiation therapy. The median PSA value at the time of protocol enrollment was 8.2 ng/mL and the median PSA double time (PSADT) was 4.7 months. Overall, 110 (74%) of the 148 patients showed metastatic progression at a median of 19 months and metastases were detected in 68% and 84% of them at 3 and 5 years, respectively. Patients with a T3 stage and Gleason score of 7 or more at the time of diagnosis were more likely to develop metastatic disease (log-rank test; *p* = 0.006), while the type of primary therapy and the presence of nodal disease were not predictive. In another study [48], the researchers analyzed 1590 patients who developed recurrent disease, among 5277 patients who had undergone either RP (1003 patients) or EBRT (587 subjects). In the RP group, about 13% of the 1003 patients had received neo-adjuvant or adjuvant androgen deprivation therapy (ADT) and the mean time between primary treatment and recurrence was 34 months. PSA data after salvage therapy were available for 1050 patients (430 patients who had received initial EBRT and 620 patients from the RP group). In 367 patients (59.2%) in the RP group, ADT was the most common salvage treatment, followed by EBRT, which was carried out in the other 248 patients (40%). ADT was also administered to 402 subjects (93.5%) in the EBRT group. Failure after salvage therapy was defined as a PSA level of >0.2 ng/mL after treatment. Overall, 420 (68%) of the 620 patients in the RP group and 319 (74%) of the 430 patients in the EBRT group recurred after salvage therapy. Among these, 739 patients (24.8%) died. The mean survival was 81.1 months (median of 59.9 months) compared with 103.2 months (median of 101.2 months) for those who did not recur after salvage therapy. This difference in overall survival was statistically significant (*p* < 0.001). A report of a phase III randomized trial [49] presented the long-term results of immediate androgen suppression and external irradiation in patients with locally advanced prostate cancer. In a subgroup of 127 high-risk patients, eight-year clinical disease-free survival (DFS) was about 40%. The high-risk category was defined as having at least two of the following three risk factors: WHO performance status > 0, G3, and initial PSA > 10. Our patient, who shares these characteristics, is alive and in good health 114 months after RP.

### 2.6. Patient 3

#### 2.6.1. History

A 39-year-old female, in her 31st week of pregnancy, presented with a 5 cm breast nodule with inflammatory signs. From ultrasound, it was suspected to be invasive cancer; subsequent biopsy confirmed a diagnosis of invasive adeno-carcinoma. From 14 October 2015 to 18 February 2016, she received neo-adjuvant chemotherapy with four cycles of 90 mg/sqm epirubicin and 600 mg/sqm cyclophosphamide by i.v. infusion, with each cycle occurring every four weeks, followed by weekly 80 mg/sqm paclitaxel by i.v. infusion. This treatment was interrupted on 18 February 2016 because of disease progression. On 3 March 2016, she underwent a radical left mastectomy; post-operative histology showed infiltrating ductal carcinoma without lympho-vascular invasion. pT2 No Mo was the post-operative TNM classification, G3, ER+ (30%) Pr+ (10%), Ki67 90%, and HER2 negative. From 28 April 2016 to 04 July 2016, she received adjuvant chemotherapy with three cycles of CMF (600/40/600 mg/sqm on days 1, 8, and 28, by i.v. infusion) and, from 31 May 2016 to 08 July 2016, she received conventional radiotherapy on the chest wall and lymph-nodes at II–III–IV levels. From 29 July 2016 to 1 February 2017, 25 mg of exemestane daily plus an LHRH agonist every 28 days were administered. On 2 February 2017, exemestane was replaced by tamoxifen because a hormonal assessment showed the recovery of ovarian functionality. On 22 May 2017, diffusion whole body magnetic resonance imaging (MRI) was carried out at the European Institute of Oncology, in Milan, and it signaled an 8 mm nodule of the middle lung lobe, which was suspected for relapse. A 13 mm nodule in the same site of the suspected relapse was confirmed on 30 May 2017 by WB CT. The subsequent biopsy showed that the nodule had malignant characteristics. On 13 June 2017, the metastatic lung nodule was removed by atypical lung resection. The immunophenotype from post-operative histology confirmed that the primary breast cancer was the probable origin; ER, PR, and Ki67 were 30%, negative, and 50%, respectively. From 18 July 2017 to 24 August 2017, the patient received four cycles of liposomal doxorubicin. From 18 September 2017 to date, this patient received four consecutive cycles of IT twice, and the last cycle was followed by one month of rest and then four successive cycles of oral ChT with vinorelbine and capecitabine. In contrast to previous reports on the metronomic use of vinorelbine and capecitabine [50,51,52] in breast cancer patients with detectable metastatic disease, one week of ChT interruption was planned because our patient did not show clinical-radiological evidence of metastatic disease. Vinorelbine at 30 mg was given every other day, and 500 mg of capecitabine was given three times a day continuously, on days 1–14 every 21 days. On 8 October 2018 and 16 October 2019, bone scanning (BS) and WB CT, respectively, were negative for relapse and circulating CEA-TPA-CA.15.3 levels were in the normal range on 14 October 2019.

#### 2.6.2. Comment

A study [53] evaluated 103 breast cancer patients who underwent surgery for suspected pulmonary metastases. In this study, the median survival time (ST) was 29 months and the three-year survival was 43% in 14 patients with a disease-free interval (DFI) of less than 24 months. All patients had received complete resections of their lung metastases. Another study [54] reported that, in 37 recruited breast cancer patients with completely resected pulmonary metastases (27 with a single pulmonary nodule (SPN) and 10 with two or more pulmonary nodules), the actuarial five-year survival was 59%. However, in five of them with a DFI of one year or less, the mean survival was only 15 ± 3.6 months, and all patients were dead at 26 months. Before, after, or both before and after thoracotomy, all 37 patients received ChT, tamoxifen, or both. Similarly, in another series [55] of 40 patients (most of them with one or two pulmonary metastases) who had undergone complete pulmonary resection, the five-year survival rate was 35.6%, while the median DFI following complete pulmonary resection was 1.6 years. Most breast oncologists, even in SPN cases, prefer systemic medical treatment to surgery. In fact, pulmonary metastasis is considered a systemic disease, and poor survival rates have been reported for those with a DFI of <3 years or with discordance in the hormone receptor status between the primary cancer and the SPN [56,57].

### 2.7. Patient 4

#### 2.7.1. History

A 65-year-old male; was a heavy smoker from 15 or 16 to 42 years old and did not have relevant co-morbidities. When he was 58 years old, he underwent a left nephrectomy for the occasional detection of the absence of the left urinary duct. On 5 May 2010, he underwent subtotal esophagectomy with intra-thoracic esophago-plastic surgery; post-operative histology showed adeno-carcinoma of the esophageal–gastric junction, intestinal type by Lauren’s classification. pT2 N0 M0 was the post-operative TNM classification, and cells were HER2 negative. He did not receive any adjuvant therapy. This patient remained disease-free until September 2016, when he went to an emergency department because of heavy abdominal cramps. On this occasion, no pathological signs were ascertained by clinical-radiological assessment, and the patient was discharged without symptoms following anti-inflammatory and anti-spastic therapy. Two months later (in November), the same signs and sialorrhoea occurred; again, he went to the same emergency department, where WB CT showed pelvic ascites and mesenteric fat thickness with small nodules suspected for peritoneal metastases. This diagnosis was confirmed at the National Institute of Cancer in Milan, where the patient was sent for consultation. From 24 November 2016 to 1 February 2017, he received four cycles of 2000/sqm/day capecitabine given orally plus 130 mg/sqm oxaliplatin by i.v. infusion (the Xelox regimen) with complete response (CR); after that, on 1 April 2017, he started maintenance cyclic IT. This patient was disease-free at the last clinical-radiological assessment on 20 May 2019. To date, 35 months from the diagnosis of metastases, he is doing well.

#### 2.7.2. Comment

The median OS of gastrointestinal cancer patients with moderate metastatic peritoneal ascites was reported to be 9.6 and 13.5 months [58,59].

#### 2.7.3. Total Lymphocytes, T Sub-populations, Cytokines, and Growth Factors Assessed in the Peripheral Blood of the Four Patients Receiving Immune Suppression Inhibiting Therapy

During IT in all four cases, the mean basal values of IL-2 increased. In patients 4 and 2, the difference was significant (** *p* = 0.046 and * *p* = 0.049, respectively) or close to significant (** *p* = 0.083). In patient 1, it was close to significant (* *p* = 0.061 and ** *p* = 0.083). The mean total number of lymphocytes decreased in two of the four cases. The difference was close to significant in patients 1 (* *p* = 0.076, ** *p* = 0.083) and 4 (* *p* = 0.079, ** *p* = 0.083). Accordingly, CD3+, CD4+, and NK basal values decreased during IT in three (75%) of the four examined patients. In particular, in contrast to NK, for which no significant difference occurred, the difference in CD3+ was significant (* *p* = 0.040) or close to significant (* *p* = 0.083) in patient 1 and close to significant (** *p* = 0.088) in patient 4. The difference in CD4+ was close to significant in both patients 1 (* *p* = 0.053, ** *p* = 0.083) and 3 (** *p* = 0.083). Basal VEGF, IL-12, and CD19+ values decreased in three (75%) patients. The difference in VEFG was close to significant (** *p* = 0.083) in patients 1 and 3. IL-12 was significant (* *p* = 0.040) or close to significant (** *p* = 0.083) in patient 2 and it was close to significant (** *p* = 0.083) in patient 1. In patients 1 (*p* = 0.063, ** *p* = 0.083) and 4 (* *p* = 0.071, ** *p* = 0.077), the difference in CD19+ was close to significant (Table 2). No significance was found for IL-6. An increase in three patients with only one close to significant was found for IL-10 and IL-2R. Conflicting results were obtained for CD8+; Treg; CD4+45RO; IL-1R; TNF alpha; TGF beta1; FGF; EGF; IGF1; and IL-1, IL-8, and CMP1 inflammatory cytokines (see Appendix A).

### 2.8. Patient 5

#### 2.8.1. History

On 17 December 2007, a 46-year-old female underwent a radical left mastectomy; post-operative histology showed an infiltrating ductal carcinoma with microscopic lesions of ductal carcinoma in situ. 

pT2 N3a (14/28) M1 was the post-operative TNM classification, G3, ER+ (90%) Pr+ (90%), Ki67 40%, and HER2 negative. The post-operative work-up, BS and WB CT showed two metastatic lesions involving the left iliac crest and pubic symphysis, respectively, with elevated serum TPA values. From 9 January 2008 to 31 July 2008, she received salvage chemotherapy with six cycles of 5-fluorouracil-epirubicin-cyclophosphamide (FEC) (600/60/600 mg/sqm on days 1 and 28) followed by four cycles of docetaxel (80 mg/sqm every 21 days). From 1 August 2008 to 2 March 2009, she was continuously given 1 mg anastrozole pills (one pill/day). On 2 March 2009, she was recruited for a pilot study, which we have more times reported [3,4,5,6], that applied cycles of IT with beta- interferon and interleukin-2 in addition to continuous conventional salvage hormone-therapy. From April to June 2010, clinical examination, bone scanning, and subsequent abdominal CT showed a complete clinical radiological response with normal values of the serum CEA-TPA-CA15.3 tumor marker panel. To date, these normal values have been maintained during intensive post-operative monitoring. In 2015 and 2016, IT cycles were interrupted for four months; in 2017, 2018, and 2019, both IT and concomitant hormone-therapy were interrupted for 7, 6, and 6 months, respectively.

#### 2.8.2. Comment

Occasionally, metastatic breast cancer patients have been reported with more than a 10-year CR following conventional therapy [60]. However, these studies had relatively poor clinical-radiological documentation [61,62].

### 2.9. Patient 6

#### 2.9.1. History

On 16 August 2011, a 64-year-old male physician with no relevant co-morbidity was suspected for transversal colon cancer by colonoscopy. On 26 August 2011, seven likely secondary lesions were detected by abdominal ultrasound in different hepatic segments (the biggest lesion was 42 mm in the largest diameter) and 15 ng/mL was the serum CEA level. On 29 August 2011, he underwent hemicolectomy of the transversal colon, and adeno-carcinoma, G2/G4 was found in post-operative histology. pT3 N2a (6 regional lymph-nodes involved of 22 examined) M1 was the post-operative TNM classification. Kras was wild type. From 14 October 2011 to 20 July 2012, he received 1000 sqm/day capecitabine given orally and 130 mg/sqm oxaliplatin by i.v. infusion (the Xelox regimen) on days 1–14 every 21 days (one cycle) for six cycles, plus 5 mg/kg bevacizumab by i.v. infusion every two weeks followed by maintenance therapy with capecitabine and bevacizumab. His serum CEA level, which was 52 ng/mL at the beginning of chemotherapy, decreased to 25 ng/mL in July 2012, and the WB CT showed slight decreases in all hepatic metastases. On 11 September 2012, all detectable liver metastases were surgically removed following his demand and, in October 2012, his serum CEA level was in the normal range. In December 2012, a radiological assessment documented a suspected secondary lung nodule in addition to the recovery of metastatic liver disease; the serum CEA value was 5.9 ng/mL, which increased to 66 ng/mL in January 2013. From January to April 2013, he received seven cycles of 130 mg/sqm irinotecan by i.v. infusion and 500 mg/sqm cetuximab by i.v. infusion every two weeks. On 2 July 2013, he underwent surgical removal of the left liver lobe; contemporaneously, other secondary lesions were enucleated (overall, ten liver metastases were removed); post-operative histology confirmed the secondary origin of the liver lesions, while a subsequent abdominal MRI signaled three regions with necrotic features. WB positron emission tomography (PET) was negative for residual metastatic liver disease. On 28 February 2013 and 1 July 2013, the serum CEA level was in the normal range. On 21 August 2013, thoracic CT showed an increase in the previously detected nodule placed at the inferior left lung lobe, while liver ultra-sonography showed two small hypo-echogenic areas suspected to be surgical scars. On 29 August 2013, the patient underwent atypical resection of the inferior left lung lobe and post-operative histology confirmed that it was a secondary lesion compatible with primary colorectal cancer. The serum CEA level was 9 ng/mL at the end of September/beginning of October. On 24 October 2013, WB CT signaled a new focal liver lesion that was 57 mm in the largest diameter and a 16 mm nodule at the lower-basal region of the right lung: both were suspected for relapse. Following oncologic consultation, from November 2013 to January 2014, he received cycles with FOLFIRI plus cetuximab. In January 2014, the serum CEA level was in the normal range. In March 2014, he underwent surgical removal of the liver nodule and of the other two nodules in the right lung (one of them was a pleural lesion); they were all determined to be secondary lesions in the post-operative histology. He continued to receive the same FOLFIRI treatment for a total of six cycles until June 2014. In July 2014, WB CT showed multiple new bilateral lung nodules. On 19 September 2014, he accepted the invitation to be included in an experimental clinical trial that was recruiting previously treated patients who had progressed under cetuximab after an initial benefit. He was selected because of his high MET level, as assessed in a biologic sample; therefore, he was given 120 mg of tivantinib (a MET-inhibitor) orally twice daily and cetuximab until 25 October 2016, with an initial clinical-radiological partial response (PR) followed by a stable disease as documented by WB PET carried out on 27 May 2016. In addition, WB PET showed a likely metastatic lesion of the lateral eighth right rib; therefore, from 13 June 2016 to 17 June 2016, in addition to tivantinib and cetuximab, he underwent RT at the rib lesion (25 Gy as a total dose by five fractionated doses). During treatment with tivantinib plus cetuximab, the serum CEA value, which was measured every 2–3 months, was in the normal range until 21 September 2016, when it was found to be 12 ng/mL. On 04 November 2016, WB PET documented the progression of the secondary lesion of the eighth right rib. On 2 December 2016, the patient underwent palliative surgical removal of the lateral arch of the VIII rib; post-operative histology confirmed the metastatic origin of the rib lesion with infiltration of the surgical edges. On 27 December 2016, surgical intervention for soft rib tissue radicalization and the removal of the middle lobe of the right lung was carried out; post-operative histology confirmed the metastatic origin of the residual disease. On 15 February 2017, WB CT showed multiple bilateral suspected secondary pulmonary lesions; the biggest was 7 mm in the largest diameter in the right lung. On 15 February 2017, he was given regorafenib, which he interrupted on 23 May 2017 because of strong side effects and the rising serum CEA level. On 17 August 2017, WB CT documented the progression of the disease at the level of the previously treated eighth rib lesion and the appearance of two new suspected lung lesions on the thoracic wall and lower right lobe (29 × 13 mm), respectively. The remaining multiple bilateral pulmonary lesions, the biggest of which was 8 mm in the largest diameter, were substantially stable. On 15 September 2017, WB PET signaled at least four suspected lung lesions and pathological activity of the VII, VIII, and IX right ribs. On 21 September 2017 and 20 November 2017, serum CEA values were 11.2 and 96 ng/mL, respectively. In November 2017, he again started treatment with the Xelox regimen plus cetuximab. After two cycles, oxaliplatin was interrupted because of strong side effects and he continued with capecitabine and cetuximab. On 13 February 2018, the serum CEA value was 46.3 ng/mL. In March 2018, WB PET confirmed the progression of metastatic disease in the right lung and right chest wall; on 20 March 2018, following transient ChT interruption, the patient underwent surgical removal of the lower right pulmonary lobe, multiple resections of the residual left lung, and thoracotomy of the VII, VIII, and IX right ribs with subsequent plastic surgery of the thoracic wall. The control with WB CT, which was carried out soon after surgery, showed that one metastatic lesion remained on the right lobe, in addition to the multiple repetitions of the left lung lobe. He started with capecitabine and cetuximab therapy, which he continued until March 2019. Serum CEA values, which were 4.1, 4.5, 3.6, 4, and 5.1 ng/mL on 6 April 2018, 24 May 2018, 1 August 2018, 10 October 2018, and 27 October 2018, respectively, increased to 12 ng/mL in December 2018. WB CT carried out in January 2019 signaled three new hepatic repetitions, the largest of which was about 2 cm, and two of them were close to the hepatic and cava veins, respectively, while the multiple lung repetitions were stable. At the beginning of April 2019, he underwent surgical removal of the hepatic lesions. Yet another WB CT carried out soon after surgery showed the persistence of metastatic liver disease in addition to the stable multiple lung repetitions. At the beginning of July 2019, he entered an experimental protocol with temozolomide that was interrupted at the end of August after two cycles because of the unresponsive metastatic disease. On 16 August, the serum CEA value was 86 ng/mL. On 2 September, following an oncologic consultation, he was given tas-102 therapy, which is ongoing. Although this patient is not disease-free, he is doing well with 0-1 ECOG PS 98 months after primary surgery.

#### 2.9.2. Comment

This patient underwent an uncommon number of surgical interventions (nine) and, during a prolonged time of post-operative monitoring, serum CEA was often in the normal range. Therefore, in this patient, surgery, in addition to a systemic treatment, likely allowed for the maintenance of an undetectable or detectable, but relatively low metastatic tumor burden. Five-year OS rates of 30%–43% were reported in patients who were submitted to complete pulmonary metastasectomy following hepatic resection for colorectal metastases at an earlier stage [63]. Similarly, a 30% five-year OS rate and a median OS ranging from 29 to 40.5 months was reported in a study [64] in which 206 inoperable patients with pulmonary oligo-metastases were treated with stereotactic body radiotherapy (SBRT). In 118 (57%) of these 206 patients, colorectal carcinoma was the primary site of pulmonary oligo-metastases. Moreover, Marudanayagam et al. [65] reported a 53% five-year OS after combined liver and lung resections for patients in which the pulmonary metastases developed more than six months after liver resection (metachronous group). However, 0% was the four-year OS for those in which, as in our patient, lung metastasis developed within six months after liver resection (synchronous group).

## 3. Discussion

In addition to those we have previously reported [2], further experimental and clinical data support the hypothesis of a close relationship between tumor growth and immune evasion in different types of cancer. Among the former, the constitutive activation of STAT3 in diverse human cancer cells and Wnt/beta catenin signaling must be mentioned. Activated STAT3 not only up-regulates genes that are critical for survival and proliferation, but also promotes the expression of immune-suppressive factors and inhibits the TH1 immuno-stimulatory molecule [66]. Wnt/beta catenin signaling is implicated in about 90% of colorectal cancers, in which it plays a role in tumor progression through mechanisms that include evasion of the immune system [67]. The relationship between tumor growth and immune evasion has been evidenced in an orthotopic murine glioma model [68], in clones from the murine epithelial ovarian cancer cell line ID18 [69], and in circulating epithelial tumor cells (CTCs). In CTCs, B7-H3 (an important immune checkpoint member of the B7 family that inhibits T-cell mediated immunity) was correlated with the proliferation marker Ki67 and was associated with the aggressiveness of tumors in breast cancer patients [70]. In another study [71], a phosphatase and tensin homolog (PTEN)-knockout murine prostate cancer-derived cell line was generated to explore genes with altered expression in PTEN-deficient cells. In this study, PTEN deficiency mobilized a variety of genes that are critical for cancer cell survival, progression, and host immune evasion. Clinical studies have reported better outcomes in metastatic patients who were in complete or partial response, compared with those with stable disease (SD) at the beginning of the maintenance therapy following conventional anti-proliferative therapy. In particular, there are studies in which patients who partially responded to systemic treatment [72] and more complete responders [73] benefitted more from maintenance therapy than patients with SD. Accordingly, PFS and/or OS in studies [74] in which complete and/or partial responders prevailed were more prolonged than in studies [75,76] in which patients with stable disease prevailed among those recruited for maintenance therapy. This supports the hypothesis of a correlation between tumour burden and immune suppression. The crucial role that MDSCs and Treg with other immune-regulatory cells play in cancerogenesis has been well described in immune-editing theory [77]. Further, it is well known that many mechanisms of immune evasion are promoted by MDSCs and Treg cells during cancer progression. A review article reported [77] that tumor-infiltrating myeloid cells (TIMs) play a key role in cancer progression. In fact, “instructed by tumor cells, myeloid cells aid in creating a tumor microenvironment (TME) that is characterized by chronic inflammation, immune-suppression and continuously proliferating tumor cells”. From these experimental and clinical observations, we hypothesized that, in the six cases reported herein, with prolonged quiescent (G0–G1 state, case 4) and/or undetectable residual cancer cells (cases 1–3, 5, and 6), it was more likely to obtain efficacious active immune manipulation and/or a spontaneous recovery of innate and acquired immunities. To date, in addition to being alive and well, all six reported cases, except for the last one, are unexpectedly free of any detectable recurrence (Table 3). In case report 1, at the last observation, more than 5 years after EBRT and 16 years after RP and BCR, serum PSA was almost undetectable. Thus, the outcome of this patient in terms of BCR and metastatic-free survival (see comment for patient 1) suggests an important slow down of biochemical and clinical progression. It is likely that a long-term permanent cell cycle arrest that was induced before apoptosis by RT [46] allowed cyclic IT to more efficaciously contrast the residual radio-resistant cancer stem cells [78]. Moreover, dose-dependent full-blown immunogenic cancer cell death (ICD) by RT and an increase in RT efficacy by TIMs depletion have been reported [77]. In the other prostate cancer patient (case report 2), BCR occurred 33 months after RP. In spite of this, on 10 October 2019, 114 months after RP, the PSA value was 0.04 ng/mL. In this patient, IAD therapy with bicalutamide alone or with an LHRH agonist and overall IT were administered for 36 and 29 months, respectively, and, as expected, during IAD therapy, serum PSA levels decreased. After BCR, from 15 May 2015 to 6 October 2015, he was under no adjuvant therapy and the mean serum daily PSA increase was 0.018 ng/mL. From 6 October 2015 to 5 October 2017, he was given IT for the first time. The mean serum daily PSA increase was 0.0054 ng/mL, about three times lower than 0.018 ng/mL. From 14 May 2018 to 7 July 2018, he did not correctly take IT and the mean daily PSA increase raised to 0.0054 ng/mL, three times higher than 0.018 ng/mL. From 22 July 2018 to 11 May 2019, he correctly received IT cycles and the mean serum daily PSA increase was 0.0013 ng/mL, about 4.5 times lower than 0.0054 ng/mL. From 10 August 2019 to 10 October 2019, he was again given cycles of IT and the mean daily PSA increase was 0.00065 ng/mL, 2 and 2.77 times lower than 0.013 ng/mL and 0.018 ng/mL, respectively. These findings point to an important slow down of BCR by cyclic IT when it was correctly administered. In cases 3 and 4, cyclic maintenance IT was administered after the removal of the metastatic lung nodule and after CR to conventional ChT, respectively. The disease-free intervals in these patients, of 28 or 32 months, suggest a significant role in the maintenance IT. In case 3, because of the aggressive and rapidly progressive disease, cycles of IT consistent with our refined model [2] and previous report [3] were alternated with cycles of conventional ChT. In these four examined patients, despite the limitation of having few available blood samples (an update will be made as soon as possible), the following hypotheses from laboratory tests can be inferred. The IL-2 increase during IT in all patients suggests an IL-2-mediated stimulation of cellular immunity, while the decrease in the total number of lymphocytes in two patients, concomitant with that of CD3+ and CD4+ cells and IL-12 [79,80,81], is consistent with their contribution to the immune response in the site/(s) of undetectable metastatic disease. Regarding the decreased basal VEGF, one should keep in mind that VEGF, in addition to stimulating angiogenesis, is a relevant immune-suppressive factor [82]. The decrease in the CD19+ basal values suggests a concomitant reduction in humoral immunity.

Case report 5 took part in a pilot clinical trial, for which we have reported many times [4,5,6,7,8]. She was one of the metastatic patients surviving in CR for more than 10 years (cumulative survival at 10 years + SE: 0.15 + 0.06) [8]. However, in contrast to all others who were given salvage hormone treatment and immunotherapy until progression, in this patient, during the last four years, both conventional salvage hormone therapy and concomitant IT were interrupted for 19 and 27 months, respectively. In spite of this, at the last WB CT assessment, the patient continued to remain in CR. This case suggests that, in some patients with prolonged CR after conventional salvage therapy, an induced spontaneous recovery of innate and adaptive immune response likely occurs [2]. Therefore, in the five patients who cyclically received the above or previously described [4] immunotherapy, active immune manipulation likely played a role in their favorable outcomes. The prolonged OS observed in the last case report could be explained by the indolent characteristic of the primary colorectal cancer. In fact, in histology, its grading was G2/G4. However, secondary liver involvement was synchronous with primary cancer, although only one month occurred between the first clinical sign and the radiological diagnosis of colorectal cancer. Furthermore, the repeated recurrences (liver, lung, and bone) were detected within a few months following the multiple radical resections that he underwent. All of these observations support the importance of maintaining an undetectable or low metastatic tumor burden over time, as documented by the prolonged relatively low serum CEA values. In this patient, it is possible that a concomitant less immune-suppressed tumor microenvironment favored a better outcome of conventional systemic therapy. In conclusion, we present a few case reports on different, high-risk solid cancers with unexpectedly favorable outcomes. According to our recently refined patho-biological model, five of the described patients were treated by targeting immune evasion during a minimal residual disease following conventional therapy. The last patient likely benefitted from maintaining a low metastatic tumor burden by surgical interventions. Overall, our case reports suggest that pharmacological immune manipulation during minimal residual disease or prolonged low tumor burden following conventional therapy produced a significant clinical benefit.

## Figures and Tables

**Table 1 ijms-20-05986-t001:** A new oral immunotherapy schedule in four advanced cancer patients with minimal residual disease.

Drug	Dose	Days
dexamethasone	1.5 mg twice a day	1 (week 1)
cyclophosphamide	50 mg per day	1–7 (week 1)
celecoxib	400 mg twice a day	8–14 (week 2)
retinyl palmitate plus dl-alpha tocopheryl acetate	30,000 IU plus 70 mg three times per day	15–28 (weeks 3–4)

Four weeks of treatment are followed by two weeks of rest; also see text.

**Table 2 ijms-20-05986-t002:** Main findings from immunological assessment in the four patients receiving inhibiting immune-suppression therapy.

**Patient 1**									
**Parameter**	**Time**	**D**	**Mean**	**SD**	***p*-Value ***	**Median**	**Minimum**	**Maximum**	***p*-Value ****
Ly tot (n)	Basal	2	1867.0	23.3	0.076	1867.5	1851.0	1884.0	0.083
	IT	3	1616	125.0		1612.0	1494.0	1744.0	
CD3+(n)	Basal	2	1596.5	6.4	**0.040**	1596.5	1592.0	1601.0	0.083
	IT	3	1383.3	82.0		1386.0	1300.0	1464.0	
CD4+(n)	Basal	2	887.0	50.9	0.053	887.0	851.0	923.0	0.083
	IT	3	725.7	59.7		715.0	672.0	790.0	
CD19+(n)	Basal	2	56.0	0.7	0.063	56.0	55.5	56.5	0.083
	IT	3	48.3	3.5		48.0	45.0	52.0	
sIL-2 (pg/mL)	Basal	2	90.2	4.2	0.061	90.2	87.2	93.1	0.083
	IT	3	122.1	14.3		127.7	105.8	132.7	
IL-12 (pg/mL)	Basal	2	64.50	7.78	0.106	64.50	59.00	70.00	0.083
	IT	3	49.00	7.21		51.00	41.00	55.00	
VEGF-A (pg/mL)	Basal	2	334.5	84.1	0.101	334.5	275.0	394.0	0.083
	IT	3	158.7	80.6		155.0	80.0	241.0	
**Patient 2**									
**Parameter**	**Time**	**D**	**Mean**	**SD**	***p*-Value ***	**Median**	**Minimum**	**Maximum**	***p*-Value ****
sIL-2 (pg/mL)	Basal	2	17.0	6.7	**0.049**	17.0	12.2	21.7	0.083
	IT	3	63.3	18.8		68.9	42.3	78.7	
IL-12 (pg/mL)	Basal	2	171.50	3.54	**0.040**	171.50	169.00	174.00	0.083
	IT	3	158.00	4.58		157.00	154.00	163.00	
**Patient 3**									
**Parameter**	**Time**	**D**	**Mean**	**SD**	***p*-Value ***	**Median**	**Minimum**	**Maximum**	***p*-Value ****
CD4+(n)	Basal	2	614.1	2.8	0.198	614.00	612.00	616.00	0.083
	IT	3	588.6	20.5		599.00	565.00	602.00	
VEGF-A (pg/mL)	Basal	2	12.00	2.83	0.247	12.00	10.00	14.00	0.083
	IT	3	45.00	30.81		33.00	22.00	80.00	
**Patient 4**									
**Parameter**	**Time**	**D**	**Mean**	**SD**	***p*-Value ***	**Median**	**Minimum**	**Maximum**	***p*-Value ****
Ly tot(n)	Basal	2	1425.0	204.0	0.079	1425.0	1281.0	1570.0	0.083
	IT	3	1035.0	139.0		987.0	926.0	1191.0	
CD3+(n)	Basal	3	917.0	134.2	0.088	994.0	762.0	995.0	0.127
	IT	3	709.3	87.5		667.0	651.0	810.0	
CD8+(n)	Basal	3	239.3	88.8	0.078	257.0	143.0	318.0	0.077
	IT	3	111.7	30.1		109.0	83.0	143.0	
CD19+(n)	Basal	3	108.8	22.7	0.071	118.0	83.0	125.5	0.077
	IT	3	75.3	7.1		74.0	69.0	83.0	
sIL-2 (pg/mL)	Basal	3	0.9	1.5	0.145	nd	nd	2.6	**0.046**
	IT	3	64.5	47.2		79.2	11.7	102.7	

D: number of determinations; IT: inhibiting immune-suppression therapy; Ly tot: total lymphocytes; CD3+, CD4+: T subsets; CD19+: B lymphocytes; IL: interleukin; VEGF-A: vascular endothelial growth factor A; nd: not detectable. * *t*-test; ** Mann–Whitney test.

**Table 3 ijms-20-05986-t003:** Outcome expected after conventional therapy and observed after addition of unconventional therapy to conventional therapy in six high risk cancer patients.

Patient (n)	Cancer Type	Non-Conventional Therapy	Expected Outcome				Observed Outcome		
			sPSA Nadir after srt (Months)	DFS (PFS)	OS (Months)	Ref.	sPSA Nadir after sRT (Months)	DFS (Months)	OS (Months)
1	Prostate	IT	10–16 12–24	<27.5 mean/median	-	43, 46	62	^b^ 201	^b^ 201
2	Prostate	IT	-	^b^ 32% at 3 years ^b^ 16% at 5 years	^b^ 59 median	47.48	-	^b^ 78	^b^ 78
3	Breast	IT ^a^	-	^c^ 19 months median	^c^ 15 ± 3.6 mean	55, 54	-	^c^ 28	^c^ 28
4	Gastric–esophageal junction	IT	-	^d^ 3.2 months (PFS)	^d^ 9.6–13.5 median	58, 59	-	^d^ 35	^d^ 35
5	Breast	Beta-IFN plus IL-2	-	-	^e^ 24–31 median	8	-	-	^e^ 140 ^e^* 112
6	Colon	Repeated surgical removals of metastases	-	-	29–40.5 median (0% at 4 years)	64, 65	-	-	^e^ 98

(also see text) IT: inhibiting immune-suppression therapy; ^a^ IT given alternately to three cycles of oral chemotherapy (also see text); sPSA: serum PSA; sRT: salvage radiotherapy; DFS: disease-free survival; PFS: progression-free survival; OS: overall survival; ^b^ from biochemical recurrence; ^c^ from surgical removal of lung metastasis (patient 3) or from primary surgery (patient 5); ^d^ from diagnosis of ascites and peritoneal metastases; ^e^ from diagnosis of metastases (32 months after complete response); ^e^* from complete response during hormone-immunotherapy; Ref.: reference number; IL-2: interleukin-2.

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
