# Peer review of "Treatment of Metastatic or High-Risk Solid Cancer Patients by Targeting the Immune System and/or Tumor Burden: Six Cases Reports"

_ijms, 2019, doi:10.3390/ijms20235986_

Round 1
Reviewer 1 Report
The present case report describes the six cancer patients with their historical treatments for approximate 10 years.
They are subjected to the pharmacological treatments and the traced measurements suggested the appreciable outcomes.
The positive parameters can be documented in any mediaum to recognize the coutcomes. However, the number of the patients is very limited in this wotk to justify their claims. Presumably, the subclassified cancer types can be repeated with the increased number for better generalization.
Neverthless, this case report contains a valuable information on the clinical tracement of such malignant or solid tumor paptients.
In conclusion, from the above circumsance, the present reviewer is flexible for documentation in IJMS ot any equivalent if the journal policy is stringent.
Author Response
We thanks the reviewers for the helpful comments.
This reviewer suggests to improve the conclusions by underlining the low number of the studied patients. Accordingly, we changed the conclusions in the abstract (last lines) and in the text (page 19, last lines).
In the text of the revised version, the extension to the entire category of patients with advanced solid cancers has been removed.
Reviewer 2 Report
The authors present 6 cases of patients with different cancer types and their responses to immunotherapy. The rationale behind combining different agents that inhibit immune suppression or promote tumor control has been explained with sufficient references to preclinical murine models. The responses of the patients have been assessed both through a detailed clinical-radiological assessment as well as evaluation of serum biomarkers unique to a cancer type, hence making the conclusions convincing. They observe that patients with either a partial or complete response or decreased tumor burden have a better immune response and sensitivity to therapy compared to those with stable disease or greater tumor burden. Increases in IL-2 and decrease in VEGF also hint at one mechanism of action of the IT being boosting anti-tumor immunity.
The report is well written with sufficient background and reference material. The history of each patient has been provided in detail and the rationale for utilizing specific dosing regimens is valid. Follow up assessment and measurement of immune parameters lend credence to the conclusion that minimal residual disease favors better immunity and response to therapy and this can be a stratification approach to choose patients likely to benefit from this therapeutic regimen.
Author Response
We thanks the reviewers for the helpful comments.
Reviewer 2 has no objection.
Round 2
Reviewer 1 Report
The case study contains the limited number of the patient. However, the clinical documentation will be valusble for the fields.
If the journal's policy support the case study, the reviewer recommend the publication in IJMS.